# The Trickle-Down Effects of Supervisor Regulatory Foci on Newcomer Task Performance

**DOI:** 10.3390/bs15020188

**Published:** 2025-02-11

**Authors:** Junzhe Zhao, Wenfan Chao, Hang Zhang, Guoxiang Zhao, Minghui Wang

**Affiliations:** School of Psychology, Henan University, Kaifeng 475001, China; zhaojunzhe@henu.edu.cn (J.Z.); wenfan@henu.edu.cn (W.C.); zhzhanghang@henu.edu.cn (H.Z.); zhaogx@henu.edu.cn (G.Z.)

**Keywords:** supervisor regulatory foci, supervisor developmental feedback, newcomer task performance

## Abstract

Supervisors’ chronic regulatory foci significantly influence their leadership styles and behaviors, with prevention focus and promotion focus exerting distinct impacts on their actions and outcomes. Drawing on regulatory focus theory, we propose a conceptual model that links supervisor promotion focus and supervisor prevention focus to new employee task performance through the mediating role of supervisor developmental feedback. We conducted a matched questionnaire survey involving 253 supervisor–new employee pairs at two time points. The findings indicated that supervisor promotion focus was positively associated with supervisor developmental feedback, whereas supervisor prevention focus was negatively associated with supervisor developmental feedback. Furthermore, supervisor developmental feedback enhanced new employees’ task performance. This study elucidates the trickle-down effects of supervisor promotion focus and supervisor prevention focus and offers practical implications for organizations on effectively managing supervisors with varying regulatory foci.

## 1. Introduction

Leadership is pivotal to organizational growth and success, warranting substantial attention and effort. [34] ([34]) have highlighted the role of leaders’ goal pursuit and social influence in effective leadership. In particular, supervisor behaviors serve as a strategy in leaders’ goal pursuit ([36]), and social influence reflects their impact on followers ([19]). Understanding these aspects uncovers the goal pursuit and social influence processes that are vital to effective leadership ([34]). Despite its significant influence, current research has insufficiently addressed these dual aspects of leadership. To bridge this gap, this study introduces the regulatory focus theory to elucidate how supervisors’ chronic regulatory foci influence their behaviors (i.e., goal pursuit) and their follower behaviors (i.e., social influence) ([17]; [34]).

Chronic regulatory foci encapsulate supervisors’ varying needs, goals, and psychological states, categorized into supervisor prevention focus and supervisor promotion focus ([4]; [10]). Supervisors with promotion focus strive for ideal self, growth, and self-realization through proactive and enterprising approaches. Conversely, those with prevention focus adopt conservative and cautious strategies to fulfill the needs of their ought self, emphasizing security and stability ([12], [13]). Therefore, supervisors characterized by different chronic regulatory foci are driven by distinct goals and motivations to exhibit different supervisor behaviors ([20]). This study adopts a “supervisor-centered” perspective to examine the effects of supervisors’ chronic regulatory foci on both supervisors and new employees. By linking supervisors’ chronic regulatory foci to specific supervisor behaviors, we aim to uncover the motivations behind supervisors’ behavior ([34]). Given supervisors’ superior expertise and skills, they often possess more informational advantages over new employees. Supervisor developmental feedback is a critical behavior that reflects the extent to which supervisors offer useful or valuable information to new employees, enabling them to learn, develop, and improve at work ([46]). Specifically, supervisors with promotion focus are more likely to encourage employee’s development, promotion, and change by providing developmental feedback ([43]). In contrast, prevention-focused supervisors tend to maintain stability by reminding employees of standards and requirements ([20]). Furthermore, promotion-focused supervisors are more likely to use a positive framework to convey information, whereas prevention-focused supervisors are more inclined to use a negative framework to convey information ([10]). Therefore, this study posits that promotion focus facilitates supervisor developmental feedback, while prevention focus hinders it.

Although valuable research exists on supervisors’ chronic regulatory foci, there are still some areas that need continued attention ([31]). One of the areas is the context-sensitivity of chronic regulatory foci, which examines specific conceptual effects in particular situations. The leadership processes driven by regulatory foci are dynamic and evolve throughout context ([21]). Specifically, supervisor promotion focus and prevention focus may play different roles in different situations. Consequently, this study investigates the impacts of supervisors’ chronic regulatory foci within the context of organizational socialization, driven by both theoretical and practical needs. From a practical point of view, the integration of new employees is closely related to organizational recruitment costs and talent retention ([27]). Therefore, effectively assisting new employees in adapting to their new environment and achieving optimal work attitudes swiftly is a growing concern for managers. In terms of theoretical needs, new employees often face more uncertainty and stress during onboarding, and the supervisor behavior is a significant factor in shaping their behaviors ([6]; [44]). [20] ([20]) developed a theoretical framework based on the regulatory focus theory and leadership theory to explain the trickle-down effect of supervisor chronic regulatory foci on employee outcomes through their behavior. However, the proposed framework has not been fully validated among new employees and has only revealed that supervisor regulatory foci can activate new employees’ regulatory foci through certain behaviors ([19]). Therefore, it is theoretically necessary to validate and further broaden the applicability of theoretical frameworks among new employees. Furthermore, research on supervisors’ chronic regulatory foci should not only address how it affects supervisors’ choices and decision-making but also explore how these decisions impact new employees managed by them ([21]). Focusing on this process helps explain the second important component of leadership, social influence ([34]). Supervisors’ chronic regulatory foci are critical determinants of their attitudes and behaviors toward new employees, subsequently affecting employees’ goal orientation, effort, and performance ([18]; [30]). Therefore, this study extends the trickle-down effect of supervisor regulatory foci to new employees’ behavior. In the process of influencing new employees through supervisor developmental feedback, supervisors’ chronic regulatory foci also trigger a trickle-down process affecting new employees’ cognition, emotion, and behavior ([11]; [19]). The long-term development of performance is a shared goal between organization and employees ([7]). Newcomer task performance, defined as the extent to which individuals accomplish duties and responsibilities related to work activities during the onboarding ([33]), serves as a key indicator of their transition from outsiders to organizational insiders ([47]). Therefore, this study uses task performance as the social influence of supervisors’ chronic regulatory foci on their followers, thereby extending the theoretical framework and supplementing the literature on regulatory foci ([20]). In addition, newcomer task performance is also an intuitive behavioral indicator to measure the organizational socialization outcome of new employees. It reveals that the specific trickle-down effects of supervisors’ chronic regulatory foci are also effective in the context of organizational socialization.

Based on the preceding discussion, this study develops the conceptual model of “supervisor regulatory foci → supervisor behavior → newcomer outcome” and makes three significant contributions to the literature on regulatory foci and leadership. First of all, this study enriches the existing research on regulatory foci in leadership by examining how supervisors’ chronic regulatory foci influence their behaviors ([36]). The study introduces supervisor developmental feedback as a critical component and demonstrates the distinct relationships between supervisor promotion focus and supervisor prevention focus with supervisor developmental feedback. This, in turn, highlights supervisor developmental feedback as an effective form of goal pursuit ([34]). Second, this study validates and extends the theoretical framework of supervisor regulatory foci proposed by [20] ([20]) by examining the trickle-down effects of supervisor chronic regulatory foci. These findings clarify how supervisors’ personal traits influence new employees’ task performance through shaping supervisor behaviors. Specially, supervisor regulatory foci may facilitate new employee task performance through supervisor developmental feedback, thereby revealing the social influence of supervisor development feedback on new employees ([34]). Finally, focusing on the new employee group broadens the application of the aforementioned theoretical framework. This not only extends the understanding of the influence mechanisms of supervisor chronic regulatory foci, but also provides a new perspective on organizational socialization processes for new employees ([21]). The conceptual model developed in this study is illustrated in Figure 1.

### 1.1. Regulatory Focus Theory and Leadership

Regulatory focus theory, introduced by [12] ([12], [13]), initiated a comprehensive exploration of motivational systems guiding goal pursuit, involving how individuals approach pleasure and avoid pain. Based on distinct needs, goals, and psychological situations, individuals operate under two distinct motivational systems: promotion focus and prevention focus ([4]). Promotion focus is characterized by the aggressive pursuit of growth and development needs, emphasizing ideal goal and striving for positive outcomes (gain or against non-gain). Individuals with promotion focus are primarily driven by intrinsic motivation, engaging in activities for personal fulfillment rather than obligation ([38]). Their aspirations center around achieving their ideal selves, encompassing hopes, wishes, and aspirations. In contrast, prevention focus entails a vigilant pursuit of security needs, where the primary goal related to duties, obligations, and responsibilities, with a significant emphasis on avoiding negative outcomes (non-losses or against losses) ([12], [13]). Individuals with prevention focus are predominantly motivated by external motivations, including social pressures, obligations, and social responsibilities ([1]). They tend to engage in activities out of necessity, fulfilling requirements rather than pursuing personal desires ([38]).

At present, an expanding body of research emphasizes the significance of chronic regulatory foci in the leadership process ([21]). Chronic regulatory foci serve as critical trait-based antecedents of supervisors’ preferences and decision-making, leading supervisors with different regulatory foci to exhibit varying leadership behaviors and styles ([34]; [36]). According to regulatory focus theory, supervisors’ goal-setting and pursuit strategies are shaped by distinct needs, goals, and situations ([12], [13]). Furthermore, different chronic regulatory foci influence whether supervisors concentrate their attention and efforts on promoting performance gains or avoiding losses, which in turn affects the priorities they establish for their employees ([20]; [34]). As a result, these supervisor behaviors driven by different chronic regulatory foci will impact the direction, effort, and overall performance of the team and employees under their leadership ([18]; [20]). Overall, examining leadership behaviors through the lens of regulatory focus theory further elucidates how supervisors’ traits shape their goal pursuit behaviors, ultimately influencing follower behaviors ([34]).

### 1.2. Supervisor Prevention Focus and Supervisor Developmental Feedback

Regulatory focus theory posits that supervisors with prevention focus tend to adopt a conservative leadership style, emphasizing the maintenance of the status quo and avoiding mistakes made by themselves and their employees ([21]). In this leadership context, such supervisors prioritize duties, obligations, and responsibilities, often acting as monitors rather than facilitators ([20]; [43]). To provide developmental feedback to new employees, supervisors must move beyond the traditional performance feedback framework, which can be more costly and challenging for prevention-focused supervisors. Specifically, supervisor developmental feedback is employee-centered and informational feedback aimed at conveying messages that assist new employees in their progress ([39]; [46]). These messages are often communicated in implicit and subtle ways to ensure that new employees are easily receptive ([8]). This process, which is characterized by a bottom-up approach ([40]), necessitates that supervisors gather relevant information about new employees, consider how to convey this information effectively, and provide targeted feedback ([39]). This can increase workload and require additional time and effort from the supervisor. Therefore, this study argues that prevention focus is not conducive to supervisors offering developmental feedback to new employees.

For prevention-focused supervisors, the extra costs associated with time and effort in supervisor developmental feedback exceed what is deemed necessary ([20]), thereby reducing the likelihood of providing developmental feedback. Second, prevention-focused supervisors typically exhibit “conservation” leadership styles and are hesitant to engage in non-essential activities, especially in context where losses may occur ([30]). Their conservative and cautious strategies may lead them to worry about whether the new employee will accurately interpret the feedback, thus heightening concerns about the potential losses associated with their developmental feedback ([30]; [36]). As a result, they are less inclined to step outside routines to offer developmental feedback to new employees. Finally, supervisors with prevention focus tend to adopt a short-term perspective, concentrating on immediate outcomes ([34]). However, supervisor developmental feedback is more concerned with how to promote the future growth and progress of the new employee ([46]). The mismatch between present-focused outcomes and future-focused growth also makes prevention-focused supervisors fail to offer favorable development feedback to new employees. In conclusion, we suggest that there may be a negative relationship between supervisor prevention focus and supervisor developmental feedback. Therefore, this study proposes the following hypothesis:

**H1.** *Supervisor prevention focus is negatively related with supervisor developmental feedback*.

### 1.3. Supervisor Promotion Focus and Supervisor Developmental Feedback

Regulatory focus theory posits that supervisors with promotion focus tend to adopt an aggressive leadership style, pursuing goals through exploratory behavior ([21]). In the leadership process, these supervisors prioritize development, progress, and breakthroughs, often acting as change agents ([2]; [43]). Unlike traditional performance feedback, supervisor developmental feedback is a kind of personalized feedback based on the characteristics and behaviors of each subordinate ([39]). Supervisor developmental feedback is also intrinsically future-oriented and aims to stimulate the intrinsic motivation of new employees to foster their development and progress ([46]). Based on these similar characteristics, we suggest that supervisor developmental feedback aligns with the goals of promotion-focused supervisors. Therefore, supervisor developmental feedback not only meets the needs of supervisors with promotion focus, but is also likely to act as their preferred behavior in the pursuit of goals ([5]).

First, supervisors with promotion focus tend to adopt an optimistic outlook regarding their future, emphasizing positive outcomes, benefits, and overall development ([14]). This perspective is also reflected in supervisor developmental feedback, which signals that supervisors view new employees’ future development positively. Rather than focusing solely on the positive or negative aspects of feedback, supervisors with promotion focus prioritize conveying information that promotes task improvement and future growth for new employees ([8]). This shared emphasis on development and progress enables promotion-focused supervisors to provide developmental feedback to new employees more seamlessly. Second, promotion-focused supervisors are inclined toward change and are more likely to engage in exploratory behaviors and take risks ([37]; [45]). They may feel dissatisfied with established feedback patterns and seek out more effective alternatives that align with their regulatory foci, such as developmental feedback. Third, promotion focus significantly enhances supervisors’ willingness to take risks ([9]), including the time and effort associated with providing development feedback. This willingness allows them to invest the necessary resources into supporting new employees’ growth. Finally, promotion-focused supervisors are more adept at adopting a long-term perspective, which aligns well with the future-oriented nature of supervisor developmental feedback ([8]). To sum up, this study posits that supervisors with promotion focus are more likely to provide developmental feedback to new employees, leading to the following hypothesis:

**H2.** 
*Supervisor promotion focus is positively related with supervisor developmental feedback.*


### 1.4. Supervisor Developmental Feedback and Newcomer Task Performance

Supervisors play a crucial role in the work environment, often seen as being representative of the organization. Research indicates that supervisors are well-positioned to transfer their regulatory foci attributes to new employees through various processes such as social learning and situational framing ([19]). New employees encounter various uncertainties and stressors upon entering an organization, making them more reliant on supervisors for support and more sensitive to supervisor influence ([6]). Consequently, supervisor developmental feedback, shaped by supervisor chronic regulatory foci, can trickle down the organizational hierarchy and influence new employees’ cognitive, affective, and behavioral responses ([25]; [34]). Supervisor developmental feedback refers to constructive information that helps new employees learn, develop, and improve their work ([46]). This feedback is oriented towards learning and development, aiming to foster new employees’ interest in the task itself rather than externally constraining them ([26]). Thus, such feedback inherently reflects characteristics of promotion focus, emphasizing progress and development. According to the regulatory focus theory, supervisors’ chronic regulatory foci and behaviors shape newcomers’ goal-seeking strategies and behaviors ([19]). Therefore, this study posits that supervisor developmental feedback can encourage new employees to adopt a proactive approach to acclimating to the organization, thereby enhancing their task performance.

Specifically, supervisor developmental feedback can guide new employees to focus on learning and improving their work, encourage them to adopt different perspectives, and boost their confidence in completing task ([8]). This feedback equips new employees with a clearer understanding of job content and task expectations, enhancing their organizational identification and, consequently, their performance ([44]). Moreover, the emphasis on learning and development in supervisor feedback heightens new employees’ interest in their work and intrinsic motivation. New employees with high intrinsic motivation are more focused on tasks and are likely to actively seek improvements or solutions when provided with feedback, thus enhancing their task performance ([26]). Finally, supervisor developmental feedback provides useful and quality information that helps new employees become familiar with their roles and tasks in a short period ([26]). Meanwhile, newcomer task performance benefits from the in-role expectations, out-of-role expectations, and cultural norms information provided by supervisor developmental feedback ([26]; [46]). In summary, this study proposes the following hypothesis:

**H3.** 
*Supervisor developmental feedback is positively related with newcomer task performance.*


Drawing on regulatory focus theory and the leadership literature, supervisor chronic regulatory foci serve as proximal motivational factors that directly influence supervisor developmental feedback. Such feedback can subsequently enhance the task performance of new employees through a trickle-down process. Based on the preceding discussion and the proposed hypotheses, this study posits that the chronic regulatory foci of supervisors not only shape their leadership behaviors but also may subconsciously influence the task performance of new employees in their daily work ([19]). In summary, this study proposes the following hypotheses:

**H4.** 
*Supervisor prevention focus has a negative indirect effect on newcomer task performance through supervisor developmental feedback.*


**H5.** 
*Supervisor promotion focus has a positive indirect effect on newcomer task performance through supervisor developmental feedback.*


## 2. Materials and Methods

### 2.1. Participants

This research used the method of questionnaire surveying, targeting enterprises in China, primarily within the service and manufacturing industries. The study was conducted according to the guidelines of the Declaration of Helsinki and approved by the Institutional Review Board of the Henan university (Number: 20230302001) in March 2023. To ensure the authenticity of the respondents’ answers, we assured respondents that the research data would be stored on a third-party platform, inaccessible to department or team leaders. The results will be used solely for academic purposes, and participants have the right to withdraw from the study at any time if they feel uncomfortable during the survey process. In this study, data were collected from direct supervisors and new employees paired together at two time points one month apart. In addition, we screened participants to ensure that all new employees had an organizational tenure of no more than one year at the time of the survey ([41]). At time 1, the questionnaire measured variables including supervisor demographic variables, supervisor prevention focus, supervisor promotion focus, and new employees’ demographics, along with their perceived developmental feedback from supervisors. A total of 310 questionnaires were returned. At time 2, the questionnaire assessed new employees’ self-rated task performance, resulting in 253 matched questionnaires.

In the supervisor sample, 136 participants were male (53.8%) and 117 were female (46.2%), with an average age of 34.079 years (*SD* = 4.500) and an average total work tenure of 11.400 years (*SD* = 5.094). Educational qualifications included 46 participants (18.2%) with a college degree, 155 (61.3%) with a bachelor’s degree, and 52 (20.5%) with a master’s degree or higher. In the sample of new employees, 125 were male (49.4%) and 128 were female (50.6%), with an average age of 27.003 years (*SD* = 3.858) and an average total work tenure of 4.281 years (*SD* = 3.898). Educational background comprised 9 (3.6%) with a high school education or lower, 65 (25.7%) with a college degree, 162 (64.0%) with a bachelor’s degree, and 17 (6.7%) with a master’s degree or higher.

### 2.2. Measures

This study employed a translation–back translation procedure to adapt an English measurement instrument for assessing the core variables. The variables were measured using a six-point Likert scale, ranging from “1 = strongly disagree” to “6 = strongly agree”.

#### 2.2.1. Supervisor Prevention Focus

This variable was assessed using the prevention focus subscale of the regulatory focus scale developed by [28] ([28]). The prevention focus subscale comprised nine items, with items such as “I am anxious that I will fall short of my responsibilities and obligations”. The Cronbach’s alpha for this measure was 0.832.

#### 2.2.2. Supervisor Promotion Focus

This variable was assessed using the promotion focus subscale of the regulatory focus scale developed by [28] ([28]). The promotion focus subscale comprised nine items, with items such as “I typically focus on the success I hope to achieve in the future”. The Cronbach’s alpha for this measure was 0.761.

#### 2.2.3. Supervisor Developmental Feedback

This variable was measured using the scale developed by [46] ([46]). The scale comprised three items, with items such as “While giving me feedback, my supervisor focuses on helping me to learn and improve”. The Cronbach’s alpha for this measure was 0.880.

#### 2.2.4. Newcomer Task Performance

Newcomers reported their task performance using the scale developed by [42] ([42]). The scale comprised four items, with items such as “Performs tasks that are expected of me”. Previous studies have shown good reliability and validity for the four-item scale ([29]). The Cronbach’s alpha for this measure was 0.794.

#### 2.2.5. Control Variables

Previous studies have indicated that the similarities in gender, age, education, and total work tenure between supervisors and newcomers significantly influence supervisor behavior and employee performance ([3]; [32]). Therefore, gender, age, education, and the total work tenure similarities were included as control variables in this study. Consistent with prior studies, similarities in age, education, and the total work tenure were calculated as an absolute difference score. Gender similarity was operationalized as a dummy variable (1 = same gender and 0 = different gender) ([32]).

### 2.3. Data Analysis

Analyses were conducted using Amos version 24.0 and Mplus version 8.0. Before testing the hypotheses, Amos 24.0 was used to conduct a test for common method biases and confirmatory factor analysis. Mplus 8.0 was then employed to test the mediation model, and bias-corrected confidence intervals were obtained with 5000 bootstrap samples.

## 3. Results

### 3.1. Common Method Biases

To address potential common method bias, this study implemented a two-wave, two-source survey design to mitigate its influence. Statistical tests were also conducted to assess potential common method bias. First, the results of the Harman one-way test indicated that the first single factor accounted for only 17.6% of the total variance, which is below the 40% threshold. Second, the study further examined common method bias by controlling for unmeasured potential method factors (ULMCs). The common method factor was added to the four-factor model, and the results showed that the change in the model fitting index (ΔCFI = 0.008, ΔIFI = 0.008, ΔSRMR = 0.001, ΔRMSEA = 0.006) did not exceed 0.01, indicating that the overall model fit did not improve significantly. In summary, these finding suggested that common method bias is not a significant concern in this study, as detailed in Table 1.

### 3.2. Confirmatory Factor Analysis

Prior to hypotheses testing, Amos 24.0 was used to perform confirmatory factor analysis for supervisor prevention focus, supervisor promotion focus, supervisor developmental feedback, and newcomer task performance. The results were shown in Table 1. The fitting of four-factor model with actual data was the best (χ^2^ = 110.967, df = 59, CFI = 0.961, IFI = 0.962, SRMR = 0.055, RMSEA = 0.059) and was better than other alternative models. Therefore, the four variables in this study had good discriminant validity.

### 3.3. Descriptive Statistics

Table 2 shows the mean, standard deviation, reliability, and correlation coefficients for the study variables. The results indicated that supervisor prevention focus was negatively correlated with supervisor developmental feedback (*r* = −0.134, *p* = 0.033), while supervisor promotion focus was positively correlated with supervisor developmental feedback (*r* = 0.128, *p* = 0.041). Additionally, supervisor developmental feedback was positively correlated with newcomer task performance (*r* = 0.163, *p* = 0.009). These finding provided initial support for our hypotheses.

### 3.4. Hypothetical Testing

Mplus 8.0 was used to test the study’s hypotheses, and the results are shown in Table 3. Specifically, Model 2 of Table 3 demonstrates that, after controlling for supervisor–newcomer gender similarity, education similarity, age similarity, and total work tenure similarity, supervisor prevention focus had a significant negative effect on supervisor developmental feedback (*b* = −0.195, *p* = 0.035). Similarly, supervisor promotion focus had a significant positive effect on supervisor developmental feedback (*b* = 0.290, *p* = 0.009).

From Model 4 in Table 3, it is evident that supervisor developmental feedback had a significant positive effect on newcomer task performance (*b* = 0.113, *p* = 0.005). The indirect effects were further validated using bootstrap sampling 5000 times with 95% confidence intervals. The results showed that the indirect effect of supervisor prevention focus through supervisor developmental feedback on newcomer task performance was −0.022 (*SE* = 0.013), with a 95% confidence interval [−0.061, −0.004], excluding zero. The indirect effect of supervisor promotion focus through supervisor developmental feedback on newcomer task performance was 0.033 (*SE* = 0.018), with a 95% confidence interval [0.007, 0.080], excluding zero.

## 4. Discussion

Drawing on regulatory focus theory and leadership studies, this study explored how supervisor prevention focus and supervisor promotion focus affect supervisor behavior and the subsequent trickle-down effects of this behavior on new employees ([34]). The finding indicated that promotion-focused supervisors were more inclined to provide developmental feedback to new employees, whereas those with prevention focus were less likely to do so. Supervisor developmental feedback, a crucial supervisor behavior, enhanced new employees’ task performance. The theoretical and management implications are as follows:

### 4.1. Theoretical Implications

Our findings contribute to research on chronic regulatory foci and leadership studies in three key ways. First, this study highlights the effects of different chronic regulatory foci on supervisor developmental feedback from a supervisor-centered perspective, advancing the leadership literature and regulatory focus theory. Supervisors’ chronic regulatory foci significantly influence leadership styles and behaviors, attracting considerable attention in the field of leadership research ([18]). While the effects of supervisors’ chronic regulatory foci on their behaviors have been explored ([2]), empirical research remains limited. This study connects supervisor promotion focus and supervisor prevention focus with supervisor developmental feedback and reveals the goal pursuit process triggered by regulatory foci ([34]). Our findings show that supervisor promotion focus positively correlates with supervisor developmental feedback, while supervisor prevention focus negatively correlates. These contrasting effects not only emphasize the necessity of distinguishing between chronic regulatory foci, but also contribute to enriching the nomological network of leadership literature ([15]; [18]).

Moreover, the findings validate the positive effect of promotion focus, where supervisors apply “aggressive” cognitive strategies to support new employee development and provide long-term developmental feedback ([18]; [20]). This result aligns with previous research, indicating that supervisors with a promotion focus are more likely to engage in risky leadership behaviors. They will alter the traditional performance feedback to offer employees developmental feedback that is constructive and valuable (e.g., [36]; [45]). This also provides important insights into how promotion-focused supervisors achieve their own goals through feedback behaviors. The results relating to prevention focus align with regulatory focus theory, indicating that prevention-focused supervisors tend to use self-focused strategies to manage employee errors ([30]). This is because prevention-focused supervisors prioritize safety-related goals and avoid risky and challenging behaviors, thereby reducing their likelihood of providing developmental feedback to new employees ([30]; [35]). This also means that supervisor developmental feedback is not an effective strategy for prevention-focused supervisors to pursue goals. Furthermore, it is important to emphasize that these results do not completely discount the value of a prevention focus. Regulatory foci are shaped by personal traits and contextual factors ([24]). From a balanced perspective, a defensive strategy can be beneficial, such as reducing unsafe behavior among employees ([15]). Lanaj et al.’s meta-analysis ([24]) suggested that structured environments enhance the advantages of prevention-focused individuals. This research encourages a positive perspective on the prevention focus, particularly when considering specific contexts. Additionally, it deepens our understanding of how supervisor developmental feedback is generated and its relationships with supervisor regulatory foci. The distinct relationships between prevention focus, promotion focus, and supervisor developmental feedback highlight feedback as a key tool for promotion-focused supervisors to achieve goals. In summary, our findings provide a useful supplement to the research on regulatory foci within the field of leadership ([21]).

Second, the study focused on new employees and their direct supervisors, using task performance as a behavioral indicator to assess the impact of supervisor developmental feedback on new employees’ organizational socialization. Previous studies have fully demonstrated the trickle-down effects of supervisor regulatory foci on regular employees’ cognitive, emotional, and behavioral outcomes ([21]; [34]). However, it is not clear whether these findings apply to this particular group of new employees. Starting from this gap, this study verifies the trickle-down effect of supervisor regulatory foci on new employees and finds that supervisor regulatory foci improve new employees’ task performance through supervisor developmental feedback. These findings extend previous theoretical framework and address the call to explore contextual effects on regulatory foci ([20], [21]). Furthermore, task performance is an important indicator reflecting the degree of newcomers’ adaptation ([27]). The result of improving task performance enriches the knowledge of the impact of supervisors’ regulatory foci on new employees’ outcomes ([19]). Therefore, our findings further interpret the effects of chronic regulatory foci on the newcomers’ organizational socialization through the lens of the trickle-down effects.

Third, this study proposed and validated the conceptual model of “supervisor regulatory foci–supervisor developmental feedback–newcomer task performance”. This complements the theoretical framework proposed by [20] ([20]) and broadens the applicability of regulatory focus theory. Specifically, this study provides a useful supplement to the above theoretical framework in two aspects. First, the theoretical framework proposed by [20] ([20]) is validated, that is, the process of supervisor regulatory foci influences new employees’ behavior by shaping their behavior. Our study sheds further light on the supervisor behaviors that may be triggered by supervisors’ chronic regulatory foci through the incorporation of supervisor developmental feedback. Second, this study also links supervisor chronic regulatory foci with newcomers’ organizational socialization. The finding also extends the framework’s application and also provides a certain direction for future research.

### 4.2. Managerial Implications

From a practical perspective, our findings offer valuable insights into managing and assigning supervisors. First, given the effects of supervisor chronic regulatory foci in our findings, organizations should prioritize understanding and utilizing this aspect to benefit both supervisors and new employees. For example, assigning tasks like onboarding new employees to the promotion-focused supervisor can accelerate integration. Supervisors with promotion focus are willing to offer developmental feedback to new employees, including valuable information related to specific task improvement and personal growth ([46]). This approach creates a win–win situation: it enables supervisors to engage in leadership behaviors that are consistent with their personal characteristics to maintain regulatory focus, while simultaneously aiding new employees in integrating into the new organization more effectively and swiftly. Second, although chronic regulatory foci are relatively stable personality traits developed over a long period of time, situational regulatory foci are somewhat malleable and can be influenced by situational factors ([19]). Supervisor behavior is shaped by both chronic regulatory foci and situational regulatory foci ([20]). These offer organizations greater flexibility in managing supervisors with varying regulatory foci. Previous research also indicates that contextual cues can trigger specific regulatory focus and influence supervisor behaviors ([30]). Therefore, organizations can utilize contextual factors to shape or modify supervisors’ situational regulatory foci. This can be achieved through leadership training aimed at reinforcing particular behaviors, thereby enhancing supervisors’ promotion focus or prevention focus to align with the specific needs of the organization and employees. Additionally, interventions can encourage supervisors to emphasize either promotion or prevention, enabling them to adjust their regulatory foci and behaviors appropriately according to different requirements. Finally, supervisor developmental feedback is also an important topic that organizations should pay close attention to when it comes to the onboarding of new employees ([26]; [46]). The impact of supervisor developmental feedback on newcomer performance underscores the need for organizations to recognize the value of supervisor regulatory foci and behaviors in managing new employees. On the one hand, organizations should recognize this importance and encourage supervisors to engage in the information practice of providing developmental feedback to new employees. For example, supervisors should keep a close eye on new employees and give them tailored feedback on how they fit into current roles and improve task productivity. On the other hand, organizations should also provide sufficient guidance to supervisors on how to give developmental feedback to new employees. Providing excellent examples of developmental feedback can be useful in enabling supervisors to give truly effective feedback.

### 4.3. Limitations and Future Research

Our study has several limitations that future studies could address. First, although this study used a two-wave paired questionnaire to mitigate common method bias, there are still some deficiencies in the research design. The social desirability effect can influence the objectivity of employees’ self-assessed performance. It is necessary to further verify the results by using other rating methods to measure employee performance. At the same time, the cross-sectional design limits our ability to explore the temporal variability of the core variables and confirm the causal relationships between the variables ([23]; [44]). Future research could use longitudinal studies or experience sampling methods to strengthen our findings. In addition, while regulatory foci can be a chronic disposition ([12]), it can also be a psychological state influenced by situational factors ([16]; [21]). Our study only measured general disposition using an established scale ([28]). Future studies could use experimental manipulations to activate situational regulatory foci and examine whether transient regulatory foci lead to similar results. Second, this study only focused on new employees with less than one year of organizational tenure, leading to limited generalizability of the results. Future research can examine whether the trickle-down effect of supervisor regulatory foci exhibits similar patterns across different employee groups from a more nuanced perspective ([20]). Third, although this study found that prevention focus reduces supervisor developmental feedback, this result does not negate the possible positive effect of prevention focus. Some scholars suggest exploring the “optimal combination” of regulatory foci from a paradoxical perspective. Supervisors may have a cautious pioneering state, which can be careful to avoid risks and mistakes while pursuing higher goals ([43]). Therefore, future research could build on this perspective to explore the possibility of coexistence between promotion focus and prevention focus. Furthermore, although this study explores the trickle-down effects of supervisor regulatory foci on new employees’ task performance, it does not directly measure whether supervisor regulatory foci influence or shape new employees’ situational regulatory foci. Previous studies indicate a leader–follower transfer model for regulatory foci ([19]), and thus a valuable research direction would be to explore how regulatory foci are transmitted within organization and the specific mechanism involved. Fourth, although our study reveals the trickle-down mechanisms through which supervisor regulatory foci affect new employees onboarding, it does not further explore the boundary conditions for these effects. Regulatory fit theory suggests that individuals will exhibit more positive cognition and greater dedication when their regulatory foci are consistent with the environment ([15]). Therefore, future research should explore how to leverage the strengths of promotion focus and prevention focus in different work settings by identifying boundary conditions. Finally, this study lacks a detailed investigation of leadership and neglects the impact of different supervisor roles/levels on the trickle-down effect of supervisor regulatory foci ([22]). Therefore, the supervisor roles and levels should be taken into account in future research involving supervisor regulatory foci.

## Figures and Tables

**Figure 1 behavsci-15-00188-f001:**
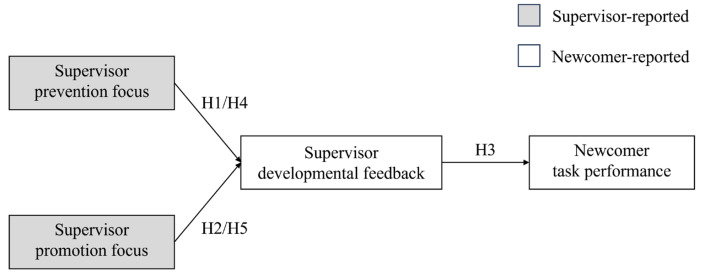
Conceptual model.

**Table 1 behavsci-15-00188-t001:** Confirmatory factor analysis results.

Model	χ^2^	df	χ^2^/df	CFI	IFI	SRMR	RMSEA
ULMC model	99.443	58	1.715	0.969	0.970	0.056	0.053
Four-factor model	110.967	59	1.881	0.961	0.962	0.055	0.059
Three-factor model	498.913	62	8.047	0.675	0.679	0.142	0.167
Two-factor model	902.337	64	14.099	0.377	0.383	0.186	0.228
One-factor model	1086.898	65	16.722	0.240	0.248	0.205	0.250

Note. N = 253. ULMC model: PROF, PREF, SDF, TP, CMV; four-factor model: PROF, PREF, SDF, TP; three-factor model: PROF + PREF, SDF, TP; two-factor model: PROF + PREF, SDF + TP; one-factor model: PROF + PREF + SDF + TP. PROF = supervisor promotion focus, PREF = supervisor prevention focus, SDF = supervisor developmental feedback, TP = newcomer task performance, CMV = common method variance, ULMC = unmeasured potential method factor.

**Table 2 behavsci-15-00188-t002:** Descriptive statistics and correlation and reliability coefficients of the variables.

Variables	1	2	3	4
1. Supervisor prevention focus	(0.832)			
2. Supervisor promotion focus	0.166 **	(0.761)		
3. Supervisor developmental feedback	−0.134 *	0.128 *	(0.880)	
4. Newcomer task performance	−0.105	0.017	0.163 **	(0.794)
*M*	3.646	4.650	4.885	4.972
*SD*	0.785	0.490	1.063	0.660

Note. *N* = 253. Cronbach’s alphas appear across the diagonal. * *p* < 0.05, ** *p* < 0.01.

**Table 3 behavsci-15-00188-t003:** Regression analysis results.

	Supervisor Developmental Feedback		Newcomer Task Performance	
	Model 1	Model 2	Model 3	Model 4
Gender similarity	0.300 *	0.300 *	−0.056	−0.091
Education similarity	0.297 **	0.243 ***	−0.045	−0.093
Age similarity	−0.000	−0.017	−0.004	−0.010
Work tenure similarity	0.017	0.035	0.002	0.006
Supervisor prevention focus		−0.195 *		−0.085
Supervisor promotion focus		0.290 **		0.035
Supervisor developmental feedback				0.113 **
*R* ^2^	0.075	0.105	0.005	0.050
Δ*R*^2^	-	0.030	-	0.045
*F*	-	4.115 *	-	3.861 *

Note. *N* = 253. * *p* < 0.05, ** *p* < 0.01, *** *p* < 0.001.

## Data Availability

Data of the present research can be available upon reasonable request.

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
