# Peer review of "The Trickle-Down Effects of Supervisor Regulatory Foci on Newcomer Task Performance"

_behavsci, 2025, doi:10.3390/bs15020188_

Round 1
Reviewer 1 Report
Comments and Suggestions for Authors
The article examines the effect of supervisor regulatory foci on new-comer task performance through the mediating role of supervisor developmental feedback.
Generally, this paper is well-organized and interesting. However, I have some comments for you to improve.
1. Collecting data from different sources is a good way to prevent CMB. However, having subordinates evaluate their performance on their own makes them question the objectivity of the assessment.
Please present all the items that measured the performance of new employees.
2. Why did you limit your target to new employees who are less than one year organizational tenure among your subordinates? Would you have achieved a different result if you expanded your target to the entire employees?
Please present more specific theories or previous studies suggestions regarding the restrictions to new employees in the research implications.
3. It is better to present the control variable before the independent variable in Table 3.
Good Luck.
Author Response
Reviewer 1
The article examines the effect of supervisor regulatory foci on new-comer task performance through the mediating role of supervisor developmental feedback.
Generally, this paper is well-organized and interesting. However, I have some comments for you to improve.
Comments 1: Collecting data from different sources is a good way to prevent CMB. However, having subordinates evaluate their performance on their own makes them question the objectivity of the assessment. Please present all the items that measured the performance of new employees.
Response 1: Thank you very much for your suggestions, which have pointed out some shortcomings and limitations in our research. Referring to previous literature, it is feasible for employees to self-evaluate their performance (Dust et al., 2022; Methot et al., 2024; Parke et al., 2018). Nevertheless, as you noted, the social desirability bias may compromise the objectivity of performance evaluations and this indeed is a deficiency in our research methodology. Therefore, we have mentioned the limitation of employee self-evaluation of performance in the discussion section and suggested that future research should use supervisor evaluations to further validate the research findings. The detailed revisions can be found on lines 525-528 of the manuscript.
Task performance was rated by new employees using 4 items from Williams and Anderson’s (1991) in-role job performance scale. The specific items are as follows:
- I adequately completes assigned duties.
- I fulfills responsibilities specified in job description
- I performs tasks that are expected of me.
- I meets formal performance requirements of the job.
Reference
Dust, S. B., Liu, H., Wang, S., & Reina, C. S. (2022). The effect of mindfulness and job demands on motivation and performance trajectories across the workweek: An entrainment theory perspective. Journal of Applied Psychology, 107(2), 221-239.
Methot, J. R., Rockmann, K. W., & Rosado-Solomon, E. H. (2024). Longing for the past: The dual effects of daily nostalgia on employee performance. Journal of Management. http://doi.org/10.1177/01492063241268695
Parke, M. R., Weinhardt, J. M., Brodsky, A., Tangirala, S., & DeV Oe, S. E. (2018). When daily planning improves employee performance: The importance of planning type, engagement, and interruptions. Journal of Applied Psychology, 103(3), 300-312.
Williams, L. J., & Anderson, S. E. (1991). Job satisfaction and organizational commitment as predictors of organizational citizenship behavior and in-role behavior. Journal of Management, 17(3), 601-617.
Comments 2: Why did you limit your target to new employees who are less than one year organizational tenure among your subordinates? Would you have achieved a different result if you expanded your target to the entire employees?
Please present more specific theories or previous studies suggestions regarding the restrictions to new employees in the research implications.
Response 2: Thank you for your constructive suggestions, which have helped us further clarify the manuscript’s logic and identify its shortcomings.
First, regarding the practical and theoretical reasons for selecting new employees with less than one year of organizational tenure as the participants:
- Practically, the onboarding of new employees is closely related to organizational recruitment costs and talent retention (Liu et al., 2024). By linking supervisor characteristics and behaviors with new employee task performance, we can not only reveal the positive impact of supervisor regulatory foci but also offer important practical advice for organizations on how to manage new employees from the supervisors’ perspective.
- Theoretically, conducting research within the group of new employees helps to validate existing research outcomes and further broaden the applicability of theoretical frameworks. Currently, a considerable amount of research has explored the trickle-down effects of supervisor regulatory foci among the general employees (e.g., Hayashi & Sasaki, 2022; Johnson et al., 2017; Kark & Van Dijk, 2019; Pearsall et al., 2023). Compared to these employees with long tenures, new employees often face more uncertainty and stress during the onboarding process, and supervisor behavior is a significant factor in shaping their behaviors (Ellis et al., 2015; Zhang et al., 2024). Therefore, it is theoretically necessary to extend the trickle-down effect of supervisor regulatory foci to the group of new employees.
In summary, our research focused on the group of new employees based on theoretical and practical needs. We have also added the above reasons for focusing on new employees in lines 68-80 of the manuscript.
Second, your query- “Would you have achieved a different result if you expanded your target to the entire employees?”- offers an interesting direction for future research. Future research can examine from a more nuanced perspective whether the trickle-down effect of supervisor regulatory foci exhibits similar patterns across different research populations. We have also added this point in lines 537-541, calling on scholars to validate the theoretical framework developed by Kark and Van Dijk (2007) across various participant groups.
Third, based on the lack of previous research, we have emphasized the research implications of studying new employees from two aspects: broadening the scope of theoretical application and enriching the content of research. The detailed revisions can be found on lines 463-471 of the manuscript.
- The study reveals that the findings on the positive impact of supervisor regulatory foci on new employees expand the applicability of theoretical framework.
- The study uses task performance to explain the impact of supervisor regulatory foci on new employees’ organizational socialization, enriching the positive outcome of the trickle-down effect.
Reference
Ellis, A. M., Bauer, T. N., Mansfield, L. R., Erdogan, B., Truxillo, D. M., & Simon, L. S. (2015). Navigating uncharted waters: Newcomer socialization through the lens of stress theory. Journal of Management, 41(1), 203–235.
Hayashi, Y., & Sasaki, H. (2022). Effect of leaders’ regulatory‐fit messages on followers’ motivation. Journal of Applied Social Psychology, 52(7), 496–510.
Johnson, R. E., King, D. D., Lin, S., Scott, B. A., Jackson Walker, E. M., & Wang, M. (2017). Regulatory focus trickle-down: How leader regulatory focus shapes follower regulatory focus and behavior. Organizational Behavior and Human Decision Processes, 140, 29–45.
Kark, R., & Van Dijk, D. (2007). Motivation to lead, motivation to follow: The role of the self-regulatory focus in leadership processes. Academy of Management Review, 32(2), 500–528.
Kark, R., & Van Dijk, D. (2019). Keep your head in the clouds and your feet on the ground: A multifocal review of leadership-followership self-regulatory focus. Academy of Management Annals, 13(2), 509–546.
Liu, S., Watts, D., Feng, J., Wu, Y., & Yin, J. (2024). Unpacking the effects of socialization programs on newcomer retention: A meta-analytic review of field experiments. Psychological Bulletin, 150(1), 1–26.
Pearsall, M. J., Christian, J. S., Burgess, R. V., & Leigh, A. (2023). Preventing success: How a prevention focus causes leaders to overrule good ideas and reduce team performance gains. Journal of Applied Psychology, 108(7), 1121–1136.
Zhang, Z., Zhang, L., Wang, H., & Zheng, J. (2024). Linking supervisor developmental feedback to in-role performance: The role of job control and perceived rapport with supervisors. Journal of Management & Organization, 30(2), 331–346.
Comments 3. It is better to present the control variable before the independent variable in Table 3.
Response 3: Thank you for your suggestion. We have made the modifications in Table 3.
Reviewer 2 Report
Comments and Suggestions for Authors
Dear Authors
It was interesting to read your article. While your writing is clear and flows well, there are areas that could benefit from significant improvement. Specifically, it would be valuable to include more recent and appropriate references to enhance the credibility and relevance of your work.
Additionally, it is important to clearly distinguish leadership within the supervisory role/level, as this has not been adequately addressed. Your discussion and conclusion sections could also be strengthened by engaging more deeply with your findings, elaborating on their implications, and clearly outlining any limitations and suggestions for future research.
Author Response
Reviewer 2
Dear Authors
It was interesting to read your article. While your writing is clear and flows well, there are areas that could benefit from significant improvement. Specifically,
Comments 1: it would be valuable to include more recent and appropriate references to enhance the credibility and relevance of your work.
Response 1: Thank you for your suggestion. We have conducted a comprehensive review of the literature to identify and incorporate more recent studies that are directly relevant to our research objectives.
Comments 2: Additionally, it is important to clearly distinguish leadership within the supervisory role/level, as this has not been adequately addressed.
Response 2: Thank you very much for your constructive suggestions, which provide an excellent direction for our future research. We did not collect detailed information about supervisor roles/levels in our data collection, which prevented us from fully exploring the impact of supervisor role/levels in the study. The issue you raised is indeed a research direction worthy of serious consideration. When collecting data that distinguishes between supervisory role/levels, in addition to the trickle-down effect from supervisors to employees, are there also other trickle-down effects between middle-level leaders and lower-level leaders/employees, as well as between top-level leaders and middle-level/low-level leaders/employees? Do these trickle-down effects exhibit similar patterns? This would be a very interesting research direction. We have emphasized this future direction in “Limitations and future research” section and also look forward to conducting related studies in the future to validate these ideas. The detailed revisions can be found on lines 560-564 of the manuscript.
Comments 3: Your discussion and conclusion sections could also be strengthened by engaging more deeply with your findings, elaborating on their implications, and clearly outlining any limitations and suggestions for future research.
Response 3: Thank you for your valuable suggestions. In the revised manuscript, we will address the following points to enhance these sections:
First, Engagement with findings. We have provided a more detailed analysis of our findings, discussing how they align with or diverge from existing literature and theories. The detailed revisions can be found on lines 423-471 of the manuscript.
- The necessity of distinguishing between two types of superior regulatory focus and their important contribution to the leadership literature network.
- Based on previous research, the specific discussions on the impacts of supervisor promotion focus and supervisor prevention focus have been respectively enhanced.
- Focusing on new employees, the discussion on the influence of supervisor regulatory foci on the newcomer task performance through supervisor developmental feedback has been strengthened.
Second, Elaborating theoretical and practical implications. We have provided more specific elaborations in the theoretical and managerial implications sections based on our findings.
- Theoretical implications. In the “Theoretical Implications” section, we elaborate on how the research findings complement the theoretical framework and provide important theoretical insights into the study of supervisor regulatory foci. The detailed revisions can be found on lines 423-471 of the manuscript.
- Practical implications. In light of the research findings, the “Managerial Implications” section has been updated with management insights related to new employees. The detailed revisions can be found on lines 513-522 of the manuscript.
Third, limitations and suggestions for future research. Incorporating the constructive advice from the other two reviewers, we have added three points regarding limitations and future research directions in the “Limitations and Future Research” section. The detailed revisions can be found on lines 525-564 of the manuscript.
- This study relies on self-reported measures from new employees to assess task performance, which may affect objectivity. Therefore, future research could validate these findings using more rigorous study designs, such as other-rating methods.
- Second, the study focusing only on new employees may lead to a limited scope of applicability for our findings, and future research can examine from a more nuanced perspective whether the trickle-down effect of supervisor regulatory foci exhibits similar patterns across different employee groups.
- Third, this study did not measure detailed information on supervisor roles/levels, which limited our ability to provide a more in-depth explanation of the impact of supervisor roles/levels on the trickle-down effect of regulatory foci. Therefore, future research on the trickle-down effects of regulatory foci should consider supervisor roles/levels as an important influencing factor.
Reviewer 3 Report
Comments and Suggestions for Authors
Thank you for allowing me to review your manuscript. This type of research is not only important but timely given the current hiring and retention challenges faced by many industries. The authors clearly stated their research focus and questions. The authors did a good job clearly identifying and stating the gaps and opportunities to expand our understanding of this research area. This demonstrated that they examined the literature and how this study can contribute to body of knowledge. The authors demonstrated that they performed a thorough review of the literature and highlighted the significance of the study. Methods were defined in detail and no issues were identified with detailed information provided on the design and sample selection. I commend them for sampling 2 groups this allows us to gain a better understanding of the impact of prevention and promotion focus. Results were explained in detail providing enough information for the reviewer to assess the results and the validity of the study. The discussion including the theoretical and practical implications provided an in-depth evaluation of the impact of this study on the body of knowledge and how it can be used to impact the workplace. The authors did a particularly good job in the discussion of the practical/managerial implications. I would have liked to see more discussion on future research maybe considering the context or country the study is performed.
I recommend that the authors review the manuscript for grammatical errors and other minor errors. For example, line 34 states "bright the gap". Did the authors mean to state "bridge this gap'? Other examples can be found in line 64, which states "researches" I am not sure if that is the correct verb. Also, see in-text citation in line 303 entry for missing ".".
Another recommendation would be to include the hypothesis in the image of the model, it would help illustrate the connections.
Author Response
Reviewer 3
Thank you for allowing me to review your manuscript. This type of research is not only important but timely given the current hiring and retention challenges faced by many industries. The authors clearly stated their research focus and questions. The authors did a good job clearly identifying and stating the gaps and opportunities to expand our understanding of this research area. This demonstrated that they examined the literature and how this study can contribute to body of knowledge. The authors demonstrated that they performed a thorough review of the literature and highlighted the significance of the study. Methods were defined in detail and no issues were identified with detailed information provided on the design and sample selection. I commend them for sampling 2 groups this allows us to gain a better understanding of the impact of prevention and promotion focus. Results were explained in detail providing enough information for the reviewer to assess the results and the validity of the study. The discussion including the theoretical and practical implications provided an in-depth evaluation of the impact of this study on the body of knowledge and how it can be used to impact the workplace. The authors did a particularly good job in the discussion of the practical/managerial implications. I would have liked to see more discussion on future research maybe considering the context or country the study is performed.
Comments 1: I recommend that the authors review the manuscript for grammatical errors and other minor errors. For example, line 34 states "bright the gap". Did the authors mean to state "bridge this gap'? Other examples can be found in line 64, which states "researches" I am not sure if that is the correct verb. Also, see in-text citation in line 303 entry for missing ".".
Response 1: Thank you for your careful review and attention to detail. We have made the corrections to the specific examples you pointed out. The authoring team conducted a line-by-line review to any other grammatical or minor errors to ensure the final document is polished and professional.
Comments 2: Another recommendation would be to include the hypothesis in the image of the model, it would help illustrate the connections.
Response 2: Thank you for your suggestion. We have included the corresponding hypothesis in the model image.